# A prospective cohort study of dynamic cell-free DNA elevation during cardiac surgery with cardiopulmonary bypass

Shlomo Yaron Ishay[1], Muhammad Abu-Tailakh[2], Lior Raichel[1], Tal F. Hershenhoren[1], Menahem Matsa[1], Oren Lev-Ran[1], Sahar Gideon[1], Amos Douvdevani[3]*

1 Soroka University Medical Center and Faculty of Health Sciences, Department of Cardiothoracic Surgery, Ben-Gurion University of the Negev, Beer Sheva, Israel, 2 Soroka University Medical Center and Faculty of Health Sciences, Nursing Research Unit, Ben-Gurion University of the Negev, Beer Sheva, Israel, 3 Soroka University Medical Center and Faculty of Health Sciences, Department of Clinical Biochemistry and Pharmacology, Ben-Gurion University of the Negev, Beer Sheva, Israel

* amosd@bgu.ac.il

**Data Availability Statement:** We have no restriction on publishing the anonymized data set of our study. Our manuscript and its supporting

## Abstract

Cardiac surgery and cardiopulmonary bypass (CPB) are associated with a systemic inflammatory reaction that occasionally induces a life-threatening organ dysfunction caused by the dysregulated host response to the damage-associated molecular patterns (DAMPs). In severe inflammation, cell-free DNA (cfDNA) and histones are released by inflammatory cells and damaged tissue and act as DAMPs. We sought to characterize the changes in circulating cell-free DNA (cfDNA) levels during CPB. Primary outcomes were renal failure, ventilation time (>18 hr), length of stay (LOS) in the intensive care unit (ICU) (>48hr), hospital LOS (>15 days), and death. We looked for associations with blood tests and comparison to standard scores. In a prospective cohort study, we enrolled 71 patients undergoing non-emergent coronary artery bypass grafting. Blood was drawn at baseline, 20 and 40 minutes on CPB, after cross-clamp removal, and 30 minutes after chest closure. cfDNA was measured by our fast fluorescent method. Baseline cfDNA levels [796 (656–1063) ng/ml] increased during surgery, peaked after cross-clamp removal [2403 (1981–3357) ng/ml] and returned to baseline at recovery. The difference in cfDNA from 20 to 40 minutes on CPB (ΔcfDNA 40–20) inversely correlated with peripheral vascular disease (PVD), longer ventilation time, and longer ICU and hospital length of stay (LOS). Receiver operating characteristic (ROC) curve of ΔcfDNA 40–20 for long ICU-LOS (>48hr) was with an area under the curve (AUC) of 0.738 ($p = 0.022$). ROC AUC of ΔcfDNA 40–20 to long Hospital LOS (>15 days) was 0.787 ($p = 0.006$). Correction for time on CPB in a multivariate logistic regression model improved ROC-AUC to 0.854 ($p = 0.003$) and suggests that ΔcfDNA 40–20 is an independent risk factor. To conclude, of measured parameters, including STS and Euro-score, the predictive power of ΔcfDNA 40–20 was the highest. Thus, measurement of ΔcfDNA 40–20 may enable early monitoring of patients at higher risk. Further studies on the mechanism behind the negative association of ΔcfDNA 40–20 with PVD and outcomes are warranted.

information file contain our minimal data set. We have uploaded our minimal dataset in Dryad: https://datadryad.org/stash/share/hY9U7stpI dGitnwD33qm1t3zPlIXyNWHSHLaDTzHqMI.

**Funding:** our work was supported by the "Dr. Montague Robin Fleisher Kidney Transplant Unit Fund (UK)." This private fund provides my department with annual support, which is not involved in our study in any way.

**Competing interests:** NO authors have competing interests

## Introduction

Cardiac surgery and cardiopulmonary bypass (CPB) are associated with a systemic inflammatory reaction that occasionally induce a life-threatening organ dysfunction caused by the dysregulated host-response to the damage-associated molecular patterns (DAMPs) [1,2], (reviewed in [3]).

Circulating cell-free DNA (cfDNA) is elevated in pathologies associated with systemic inflammatory response syndrome (SIRS), including sepsis [4,5] and severe trauma. In SIRS, a major source of cfDNA is neutrophil extracellular traps (NETs) released from activated neutrophils that undergo NETosis. NETs are DNA-based fibrous structures that trap and kill microorganisms with antimicrobial proteins [6]. When degraded, NETs are a source of circulating cfDNA and histones and act together as DAMPs to contribute to inflammation [7], thrombosis [8], and endothelial injury [9]. Growing evidence supports the hypothesis that DAMPs, including cfDNA, contribute to multiple organ dysfunction syndrome (Reviewed in [10]). Notably, high post-surgery levels of cfDNA predict renal dysfunction in patients after cardiac surgery [11].

We developed a simple fluorescent method applied directly to biological samples to measure cfDNA concentrations that could easily become part of routine testing performed in clinical laboratories and at the bedside [12]. In contrast to most other DNA assays, our fast test does not require prior processing of samples, i.e., DNA extraction and PCR amplification. It is simply performed by adding diluted fluorochrome (SYBR® Gold dye) to samples and measuring fluorescence. cfDNA measured by our method was an excellent prognostic indicator for various morbidities [4,13,14]. For example, admission cfDNA level had better discriminatory ability than the gold standard APACHE II score and procalcitonin for 28-day mortality of patients with severe sepsis [4].

Our current study is the first to monitor the dynamic changes of cfDNA levels during on-pump coronary artery bypass grafting (CABG) surgery with the aim to characterize the changes in cfDNA levels during CPB and the access to their association with outcomes compared to standard scores.

## Materials and methods

### Cohort group

This single-center prospective cohort study was conducted from July 2009 to August 2010 in the Cardiothoracic Surgery department at Soroka University Medical Center (SUMC). The study protocol was approved by the SUMC institutional review board (#4636) and registered on ClinicalTrails.gov (NCT05094960). Following explanation from authorized personnel, all participants gave their written informed consent per the Helsinki Declaration. According to regulations, signed forms are saved in the study folder. The study was carried out under relevant guidelines and rules. We enrolled 71 consecutive patients undergoing non-emergent first-time CABG. Each patient was assessed with five blood samples for cfDNA: Baseline before skin incision, 20 minutes after initiation of CPB, 40 minutes after initiation of CPB, immediately after cross-clamp removal, and during recovery, 30 minutes after chest closure.

### Sample size considerations

According to preliminary data we expected cfDNA values before surgery to average around 700±450 ng/ml and to double during bypass. Based on these assumptions and under a type I error probability of 5% ($\alpha = 0.05$), with 70 patients, our study was expected to have a statistical

power of 80% to reject the null hypothesis and identify a change of 100% difference in cfDNA levels.

## Clinical outcomes

Primary outcomes were prolonged postoperative mechanical ventilation time (>18hr), renal failure (creatinine >2), prolonged length of stay (LOS) in the intensive care unit (ICU) (>48hr), atrial fibrillation, extended hospital LOS (>15 days), and surgery-related death. Outcomes were documented following surgery up to 28 days. Secondary outcomes were the association of cfDNA with blood tests and performance in comparison to standard scores.

## Surgical technique

A heparin loading dose was administered to achieve a kaolin-activated coagulation time of 480 seconds or more. Right atrial appendage-to-ascending aorta cannulation was performed to institute CPB. Compatible with our standard technique, operations were performed during tepid cooling (32–34˚C). Distal coronary anastomoses were performed during a single aortic cross-clamp after cardioplegic arrest. Cold (10˚C) blood cardioplegic solution was delivered in a 4:1 ratio, antegrade through the aortic root. After cardioplegic induction (10 ml/kg), intermittent doses (300 to 500 ml) were administered after the completion of each distal anastomosis. Proximal anastomoses were performed during aortic cross-clamp without the use of partial (side-biting) clamps.

## Criteria for extubation

Anesthetic techniques and early extubation protocols are designed to achieve the "fast-track" of most patients.

## Weaning criteria

Awake with stimulation, Adequate reversal of neuromuscular blockade, Chest drain tube drainage < 50ml/hr, Core temperature > 35˚C, Hemodynamic stability, Satisfactory blood gases: PaO2/FiO2 >150; pCO2<50; pH– 7.3–7.5.

## Criteria to define PVD in patients

Claudication, either with exertion or at rest, amputation for arterial vascular insufficiency. Vascular reconstruction, bypass surgery, or percutaneous intervention to the extremities (excluding dialysis fistulas and vein stripping). Documented abdominal aortic aneurysm with or without repair. Documented subclavian artery stenosis. Peripheral arterial disease excludes disease in the carotid, cerebrovascular arteries, or thoracic aorta. PVD does not include deep vein thrombosis, pulmonary artery aneurysm, Raynaud's Disease, or arteriovenous malformations.

## Criteria for ICU discharge

To be discharged from the ICU patient needed to be fully conscious, cooperative, stable hemodynamically, without pressors or inotropes support, stable in its need for respiratory aid, at most with nasal oxygen cannula, does not bleed, and have normal urine output with no exceptions in laboratory tests.

### Criteria for hospital discharge

The patient's earliest discharge was on postoperative day four.

To be discharged, the patient needed to be fully conscious, cooperative, hemodynamically stable (preoperative rhythm, steady blood pressure), and respiratory stable with room air blood saturation above 92%. Normal chest x-ray, no fever, good wound healing, reasonable pain control, and no exceptions in laboratory tests.

### cfDNA measurement

Blood samples were collected in commercial gel tubes using BD Vacutainer® SST II plastic serum tubes with silica (clot activator) gel (Becton-Dickinson, Plymouth, UK). Sera were separated by centrifugation (4°C, 2000g, 10 min) and kept at -20°C until assayed. cfDNA was quantified on coded serum samples, by a rapid SYBR® Gold fluorometric assay, which does not require prior processing of samples, i.e., DNA extraction and amplification [12]. Briefly, 20μL of patient serum was applied in duplicate to black 96-well plates (Greiner Bio-One; Frickenhausen, Germany). 80μL of diluted SYBR® Gold (Invitrogen, Paisley, UK) was added to each well (final dilution 1:10,000), and fluorescence was measured with a 96-well fluorimeter (SpectraMax Paradigm plate reader, Molecular Devices, San Jose, CA) at an emission wavelength of 535 nm and an excitation wavelength of 485 nm. Concentrations of unknown samples were calculated by extrapolation in a linear regression model, from a standard curve (39–5000 ng/ml) of sonicated Salmon sperm DNA (Sigma-Aldrich).

### Statistical analysis

For continuous variables with normal distribution, patient characteristics are presented as mean ± standard deviation (SD), non-parametric variables are shown as median and 95% confidence interval (95%CI), and categorical variables are presented as percentages. Groups were compared using the chi-square (*p*-value) or variable characteristics and normal distribution t-test, non-parametric Mann-Whitney test or Wlicoxon test for matched measurements. For comparison of more than two groups, we used the Kruskal-Wallis test or Friedman test for matched measurments. For correlations, we used the non-parametric Spearman test, and the Pearson test was used for normally distributed numerical values. We used multivariate logistic regression models for the prediction of extended hospital LOS. Statistical analysis was performed using Prism 9.02 software.

## Results

Baseline characteristics are presented in Table 1. All 71 patients were non-emergent with low surgical risk (mean Society of Thoracic Surgeon (STS) score 0.75±0.4), mean age 63.1±10.34 years, mostly men (85.9%). Mean CPB time was relatively short (71.48 ± 26.63 minutes). Only four patients had cross-clamping time >100 minutes (mean cross-clamping time 66.58±27.70 minutes). Main 30 days outcomes are depicted in Table 1. There was no mortality; nine patients needed prolonged LOS in the intensive care unit (ICU) (>48hr), and nine patients required extended hospital LOS (>15 days).

As shown in Fig 1A, baseline cfDNA levels before skin incision increased during surgery and peaked after cross-clamp removal. Baseline median (95%CI) cfDNA concentration before skin incision were 796 (656–1063) ng/ml, engagement to CPB caused a gradual increase of cfDNA levels to 1308 (1113–1699) ng/ml (*p*<0.01)] at 20 minutes, 1805 (1597–2284) ng/ml (*p*<0.01) at 40 minutes and 2403 (1981–3357) ng/ml (*p*<0.01) after cross-clamp removal. cfDNA at cross-clamp removal correlated to cross-clamp time (Rho = 0.429, *p* = 0.0017 not

**Table 1. Baseline demographic and clinical parameters.**

| Baseline characteristics (N = 70) | Mean±SD or N (%) |
|---|---|
| Age (years) | 63.1±10.34 |
| Male gender | 61 (85.9) |
| Hypertension | 27 (38) |
| Peripheral vascular disease | 8 (11.3) |
| Diabetes mellitus | 28 (39.4) |
| Smokers | 20 (28.2) |
| STS score | 0.75±0.4 |
| Normal to Mild LV dysfunction (EF– 40%-70%) | 51 (71.8) |
| Moderate to severe LV dysfunction (EF≤40%) | 20 (28.2) |
| Prolonged mechanical ventilation (>18hr) | 4 (5.6) |
| Prolonged ICU stay (>48hr) | 9 (12.7) |
| Prolonged length of hospital stay (>15 days) | 9 (12.7) |
| Renal failure (Cr >2) | 5 (7) |
| Atrial fibrillation | 25 (35.2) |
| Deep sternal wound infection | 0 (0) |
| Cerebrovascular accident | 0 (0) |
| 28 days mortality | 0 (0) |

STS- Society of Thoracic Surgery, EF–Ejection fraction, LV–Left ventricle, CPB–cardiopulmonary bypass, ICU-Intensive Care Unit, Cr- creatinine.

shown). From the peak at cross-clamp removal, cfDNA levels sharply decreased to near baseline levels 669 (533–1121) ng/ml ($p<0.0001$) after 30 minutes of recovery.

We looked for correlations of primary and secondary outcomes with cfDNA levels at the various surgery time points. In addition, we correlated outcomes to the dynamic changes in cfDNA (ΔcfDNA) between time points, including cfDNA at 20 minutes minus baseline, elevation of cfDNA during 20 to 40 minutes (ΔcfDNA 40–20), ΔcfDNA cross-clamp removal -40, ΔcfDNA cross-clamp removal-recovery, ΔcfDNA cross-clamp removal-0 and ΔcfDNA0-recovery. Of these correlations, outcomes correlated best with ΔcfDNA 40–20. Selected correlations are presented in Table 2.

In patients with short ICU-LOS (<18hr) and moderate ICU-LOS (18-48hr), we observed a significant elevation of cfDNA during 20 to 40 minutes of CPB ($p<0.01$ and $p<0.0001$ accordingly). While baseline cfDNA levels in these groups were not different ($p = 0.539$), the trend of reduction in ΔcfDNA 40–20 correlated to ICU-LOS (Rho = -0.365, $p = 0.002$). In patients that needed prolonged ICU-LOS (>48hr), the elevation of ΔcfDNA 40–20 did not reach significance, and ΔcfDNA 40–20 was the lowest and significantly lower than patients with ICU-LOS <18hr ($p<0.01$, Figs 1B and 2A). We found no significant difference in cfDNA levels between the groups at 20 minutes (Fig 1B). Therefore, the reduction in ΔcfDNA 40–20 levels were caused mainly by the reduced changes at 40 minutes that differ significantly between the <18 and >48 hours groups (Fig 1B, $p<0.05$). Receiver operating characteristic (ROC) curve of ΔcfDNA 40–20 for long ICU LOS (>48hr) was significant ($p = 0.022$) with an area under the curve (AUC) of 0.738 (Fig 2B).

Similar to ΔcfDNA 40–20, baseline neutrophil numbers were not different between these three groups (not shown), and their number at recovery was lower in patients who stayed more than 48hr compared to patients released from ICU for less than 18hr (Fig 2C). Patients' ΔcfDNA 40–20 correlated to neutrophil number at recovery (r = 0.378, $p = 0.0013$, Fig 2D).

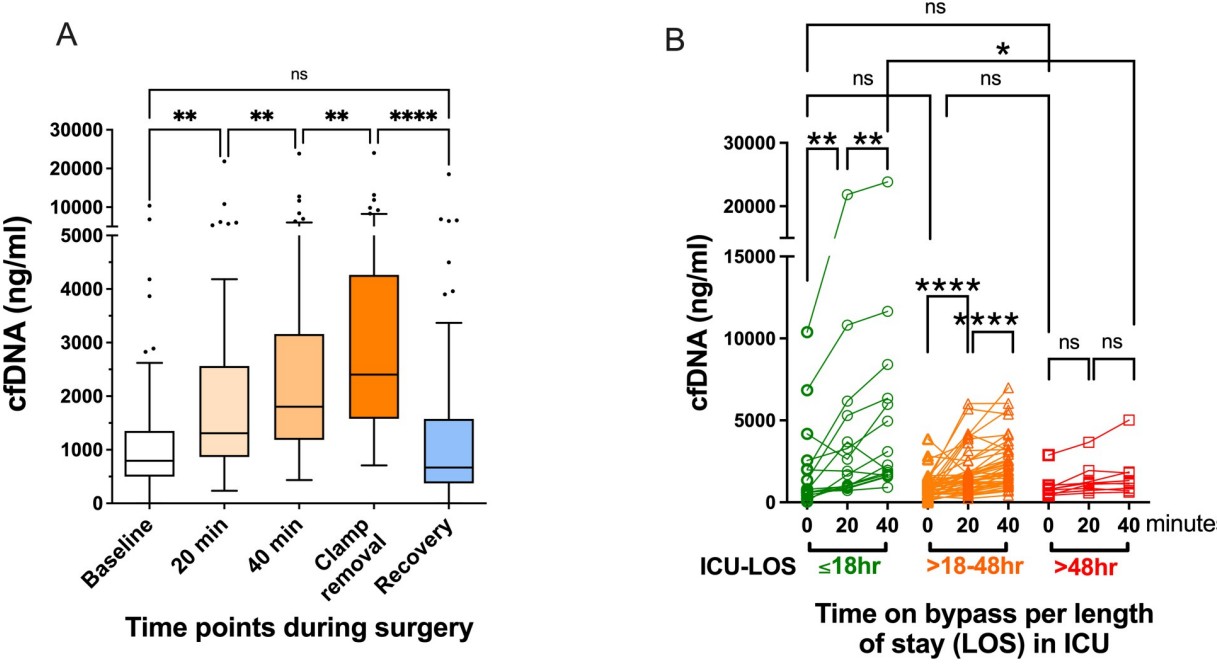

**Fig 1. Cell-Free DNA (cfDNA) according to consecutive time points of coronary artery bypass grafting surgery (CABG). A)** Blood was drawn from 71 patients undergoing CABG before skin incision (Baseline), 20 and 40 minutes after initiation of cardiopulmonary bypass (CPB), immediately after cross-clamp removal, and in recovery of 30 minutes after chest closure. cfDNA was measured by a fast fluorescence assay. **B)** cfDNA levels at 20 and 40 minutes CPB (ΔcfDNA 40–20) according to intensive care unit (ICU) length of stay (LOS). **p<0.01, ****p<0.0001, ns-not significant, tested by non-parametric Friedman test, Kruskal-Wallis test and Wilcoxon test.

Third (33%) of the patients with prolonged ICU-LOS (>48 hours) suffered from peripheral vascular disease (PVD), while in patients with shorter LOS, the PVD rate was only 9.6% (Chi-square test $p$ = 0.0251). Interestingly, of all comorbidities, ΔcfDNA 40–20 was lower in patients with PVD, 90.5 (-164-814) vs. 614 (443–751) in patients without PVD ($p$ = 0.022, Fig 2E).

Table 2 shows the correlations of selected parameters with outcomes. Prolonged ventilation time and ICU-LOS were in negative correlation to cfDNA levels at 40 minutes and ΔcfDNA 40–20. In addition, ICU-LOS negatively correlated to post-surgery neutrophils number, albumin, and cfDNA after clamp removal. The strongest correlation to ICU-LOS was the negative correlation of ΔcfDNA 40–20 (Rho = -0.365, $p$ = 0.002), slightly better than the positive correlations to Euroscore (Rho = 0.346, $p$ = 0.003) and STS score (Rho = 0.279 $p$ = 0.021). In contrast to ΔcfDNA 40–20 correlation with hospital LOS (Rho = -0.266, $p$ = 0.0263), correlations of Euroscore (Rho = 0.116, $p$ = 0.335) and STS (Rho = 0.200, $p$ = 0.104) score were not significant (Table 2 and Fig 3A).

In contrast to scores of Euroscore and STS that were indifferent to hospital LOS, the median ΔcfDNA 40–20 in patients that needed prolonged hospital LOS (>15 days) was significantly lower in patients that stayed in hospital less than eight days ($p$<0.05, Fig 3A). ROC AUC of ΔcfDNA 40–20 to predict long Hospital LOS (>15 days) was 0.787 ($p$ = 0.006). AUCs of Euroscore and STS were low and did not reach significance (0.511, $p$ = 0.917 and 0.626, $p$ = 0.229 accordingly).

As previously shown, CPB duration influences patient outcomes after cardiac surgery [15]. Similarly, in our study, time on CPB correlated to Hospital LOS (r = 0.369, $p$ = 0.007). In a multivariate logistic regression model for predicting long hospital LOS, we adjusted the effect of ΔcfDNA 40–20 with time on CPB (Table 3 and Fig 4). The odd ratio for prolonging hospital

**Table 2. Correlations of risk factors, scores, and blood tests to major outcomes.**

| Parameter | Ventilation (>18hr) Rho | p-value | Renal Failure (Cr>2) Rho | p-value | LOS in ICU (>48hr) Rho | p-value | LOS in Hospital (>15d) Rho | p-value | Atrial Fibrillation Rho | p-value |
|---|---|---|---|---|---|---|---|---|---|---|
| Age | - | ns | - | ns | - | ns | - | ns | 0.334 | 0.0044 |
| Gender (1 = Male) | - | ns | - | ns | - | ns | ns | ns | 0.295 | 0.0125 |
| BMI | - | ns | - | ns | - | ns | ns | ns | - | ns |
| HTN | - | ns | - | ns | - | ns | 0.27 | 0.0225 | - | ns |
| DM | - | ns | - | ns | - | ns | - | ns | - | ns |
| PVD | - | ns | - | ns | - | ns | ns | ns | - | ns |
| Smoke | - | ns | - | ns | - | ns | ns | ns | - | ns |
| Lymphocytes Pre | - | ns | - | ns | - | ns | - | ns | - | ns |
| Lymphocytes post | - | ns | - | ns | - | ns | - | ns | - | ns |
| Neutrophils pre | - | ns | - | ns | - | ns | - | ns | - | ns |
| Neutrophils Post | - | ns | - | ns | -0.242 | 0.042 | - | ns | - | ns |
| Neutrophil lymphocyte ratio Pre | - | ns | - | ns | 0.308 | 0.009 | - | ns | - | ns |
| Neutrophil lymphocyte ratio post | - | ns | - | ns | - | ns | - | ns | -0.337 | 0.004 |
| Hemoglobin pre | - | ns | - | ns | - | ns | - | ns | - | ns |
| Hemoglobin post | - | ns | - | ns | - | ns | - | ns | - | ns |
| Albumin pre | - | ns | - | ns | - | ns | - | ns | - | ns |
| Albumin post | - | ns | - | ns | -0.302 | 0.011 | - | ns | - | ns |
| Euroscore | - | ns | - | ns | 0.346 | 0.003 | - | ns | - | ns |
| STS | - | ns | 0.427 | 0.0003 | 0.279 | 0.021 | - | ns | 0.257 | 0.0356 |
| cfDNA Baseline | - | ns | - | ns | - | ns | - | ns | - | ns |
| cfDNA 40 min | -0.259 | 0.0304 | - | ns | -0.332 | 0.005 | - | ns | - | ns |
| CfDNA Clamp removal | - | ns | - | ns | -0.238 | 0.047 | - | ns | - | ns |
| cfDNA recovery | - | ns | - | ns | - | ns | - | ns | - | ns |
| ΔcfDNA 40–20 min | -0.299 | 0.0121 | - | ns | -0.365 | 0.002 | -0.2655 | 0.0263 | - | ns |
| ΔcfDNA Post-Pre | - | ns | - | ns | - | ns | - | ns | - | ns |

LOS of ΔcfDNA 40–20 is 0.83 ($p = 0.0475$). The ROC of this model is shown in Fig 4. The AUC of cofounder adjusted ΔcfDNA 40–20 was higher, compared to ROC without adjustment; AUC 0.854 ($p = 0.003$) vs AUC 0.787 ($p = 0.006$).

## Discussion

It is recognized that the CPB induces a systemic inflammatory response with the generation of oxygen-free radicals, cytokine release, and altered nitric oxide release, presumably due to the contact of blood with foreign surfaces [16,17]. cfDNA in disease and trauma patients depends mainly on bacterial or DAMPs load. In cardiac surgery, inflammation from CBP is similar in all patients; therefore, cfDNA between patients varies due to surgery time and individual intensity of systemic inflammatory response. In our study, we now demonstrate for the first time that during surgery, CPB is associated with a time-dependent increase of cfDNA levels with a peak at cross-clamp removal and return to basal levels shortly after chest closure.

The low complication rate and no mortalities in our cohort are explained by its composition of low-risk patients, as defined by their scores, comorbidities, and CPB time. The low-risk cohort might explain that we found no correlation between post-surgery cfDNA levels and renal dysfunction, a correlation seen in other studies [11,18]. Correlation of cfDNA with kidney injury was found in studies performed on complicated patients. For example, in the study

A

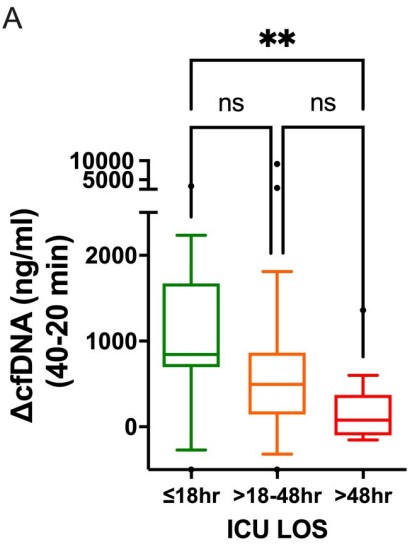

B

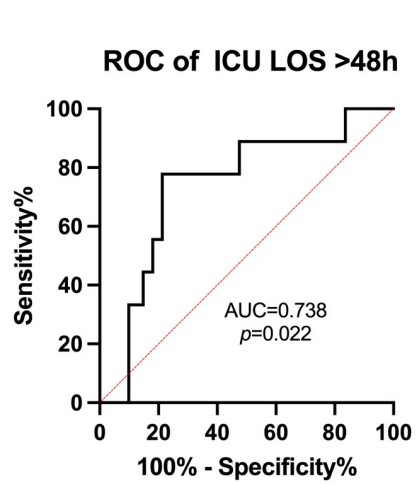

C

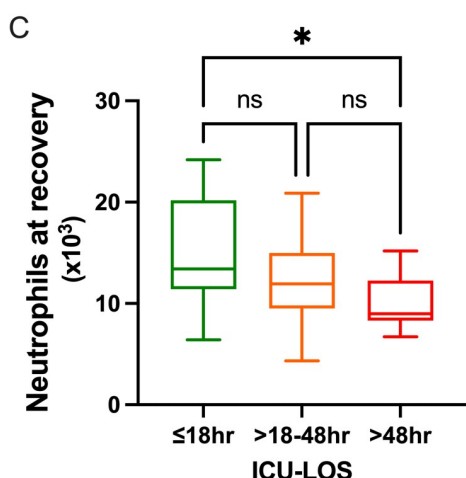

D

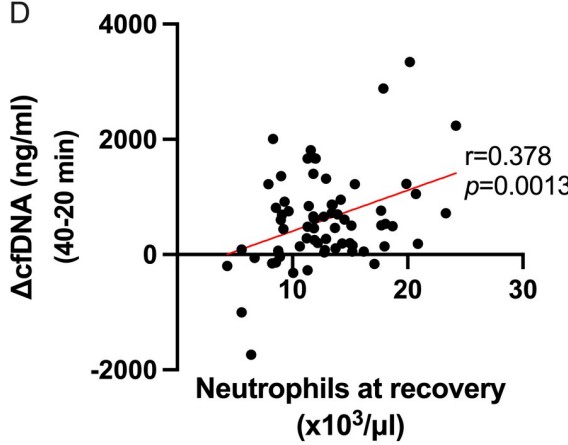

E

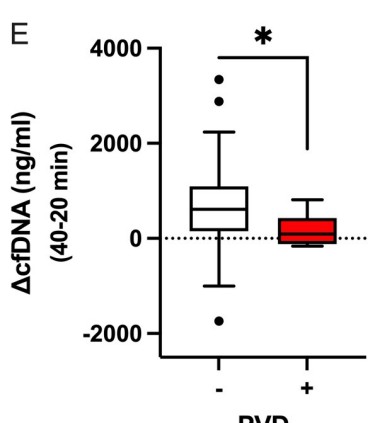

**Fig 2. Changes in cfDNA and neutrophil numbers at recovery according to the length of stay (LOS) in ICU. A)** ΔcfDNA 40–20 according to ICU LOS. **B)** ΔcfDNA 40–20 receiver operating characteristic (ROC) curve for long ICU LOS (>48hr). AUC-Area under the curve. **C)** Neutrophils numbers at recovery according to ICU LOS. *$p<0.05$, **$p<0.01$, ****$p<0.0001$, ns-not significant Kruskal-Wallis test. **D)** Linear correlation between (ΔcfDNA 40–20) to neutrophils number at recovery from surgery. ΔcfDNA 40–20 correlation to ICU-LOS, Spearman r = -0.365, $p = 0.0019$ and neutrophils correlation to ICU-LOS r = -0.243, p = 0.0415. **E)** ΔcfDNA 40–20 according to PVD. Mann-Whitney test, *$p<0.05$.

of Qi et al., the median CPB duration was 197 minutes [18], three times longer than our cohort's CPB duration. CPB duration is considered a critical factor that influences patient outcomes after cardiac surgery [15].

ΔcfDNA 40–20 was strongly associated with outcomes, more than all other parameters we tested in this study, including Euroscore and STS, the standard multifactorial prognostic scores. For prolonged ventilation, ICU-LOS and Hospital-LOS ΔcfDNA 40–20 levels were significantly lower than in patients without prolonged ventilation and short LOS. For prolonged LOS, ROC AUCs of ΔcfDNA 40–20 for ICU and hospital were 0.738 and 0.787, higher than AUCs of Euroscore and STS.

However, unlike what we expected, the dynamic changes in cfDNA levels (ΔcfDNA 40–20) were lower in patients with worse outcomes, including prolonged ventilation time and prolonged LOS in ICU and hospital. Surgery time may be too short for damaged tissue to release a considerable amount of cfDNA, and what we observe in this short time frame is primarily released from fast-reacting neutrophils. As in SIRS associated with sepsis or trauma [19,20], the primary source of cfDNA in inflammation induced by CPB could be NETs released from activated neutrophils that undergo NETosis [21–24]. Support for this possibility is the correlation of neutrophil number at the end of the surgery with ΔcfDNA 40–20. Low neutrophils number is associated with reduced elevation of cfDNA. Thus, it is possible that in patients with complications, the inverse correlation of ΔcfDNA 40–20 with outcomes is caused by impaired neutrophils that fail to recruit and release less cfDNA.

An additional conceivable explanation for the inverse correlation of ΔcfDNA 40–20 with outcomes is derived from the inverse correlation of ΔcfDNA 40–20 with PVD and the association of PVD with outcomes. As for inferior outcomes, patients with PVD had lower ΔcfDNA 40–20 than those without PVD. It is well-known that vascular health affects outcomes [25]: In our study, a third (33%) of the patients with prolonged ICU-LOS suffered from PVD, while in patients with shorter LOS, the PVD rate was only 9.6%. In CPB, it was demonstrated that despite full blood heparinization, consumptive coagulopathy related to ongoing intrinsic and extrinsic system activation and thrombin generation continues at low levels [26]. Endothelial dysfunction is associated with pro-coagulation [25], perhaps the worse outcome of patients with PVD is related to uncontrolled coagulation driven by proinflammatory cytokines and DAMPs. Coagulation in patients with vascular disease may lock neutrophils in the microvasculature and reduce cfDNA released into circulation. This suggested mechanism is attractive since it explains in a patient with complications, bleeding due to consumptive coagulopathy, and the damage to organs by capillary obstruction. If this mechanism is valid, reduced changes in cfDNA during CPB indicate an elevated risk for blood loss and organ dysfunction.

NETs and cfDNA are probably not bystander in microvascular obstruction of damaged vessels. Several studies describe the damage of NETs to endothelial cells during CPB. For example, the work of Weber et al., who show in a rat model that DNase I diminishes endothelial dysfunction and inflammation [23], and the study of Paunel-Görgülü et al. that demonstrate in patients after long CPB (>100 minutes), that the elevation of cfDNA after the end of surgery correlates to markers of endothelial damage [22]. Probably a considerable part of cfDNA elevation after surgery, reported in this paper and the work of Qi et al. [18], is the necrosis of

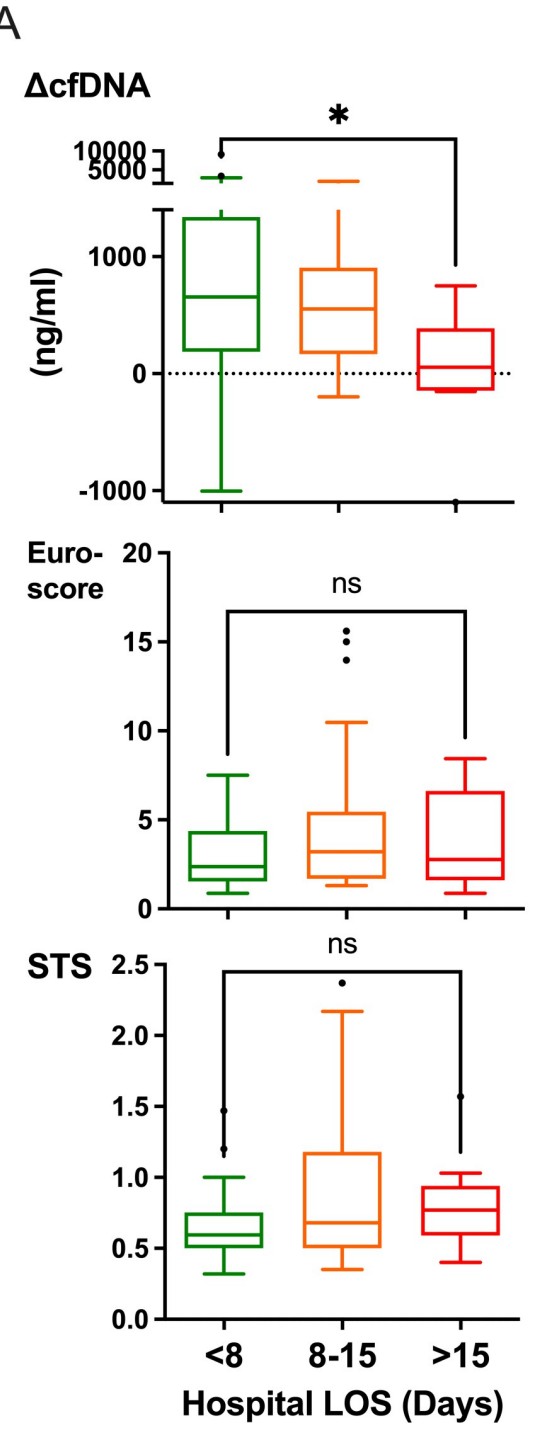

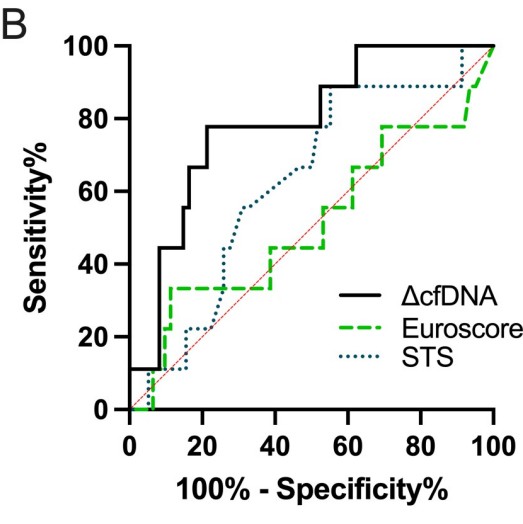

**Fig 3. ΔcfDNA and scores according to hospital length of stay (LOS), and univariate ROC curves. A**) ΔcfDNA (40–20 minutes), Euroscore and Society of Thoracic Surgeon (STS) score according to LOS. * *p*<0.05, ns-not significant, Kruskal-Wallis test. **B**) Scores univariate ROC curves for long LOS (>15 days). AUC (95% CI) for ΔcfDNA = 0.787 (0.64–0.94), *p* = 0.006, for Euroscore = 0.511 (0.29–0.74), *p* = 0.917, for STS = 0.626 (0.44–0.81) *p* = 0.229.

obstructed microvesicles and the degradation of NETs they contained. Further studies are required to depict the mechanism behind the negative association of ΔcfDNA 40–20 levels with PVD and poor outcome.

**Table 3. Logistic regression models for prediction of long hospital LOS (>15 days) by ΔcfDNA 40–20 adjusted with time on CPB.**

| Variable | B | SE | Odd ratio | 95% CI | \|Z\| | *p*-value |
|---|---|---|---|---|---|---|
| **DcfDNA 40–20** (mg/ml) | -0.183 | 0.09231 | 0.8328 | 0.6650 to 0.9671 | 1.982 | **0.0475** |
| **Time on** CPB (min) | 0.03526 | 0.01818 | 1.036 | 1.006 to 1.081 | 1.939 | 0.0525 |

Compared to Euroscore and STS, which did not predict prolonged hospital-LOS, ΔcfDNA 40–20 prediction was significant with a ROC-AUC of 0.787. Moreover, correction for time on CPB in a multivariate logistic regression model improved ROC-AUC and suggests that ΔcfDNA 40–20 is an independent risk factor The simple fluorescent assay we used to measure cfDNA [12] may enable real-time detection of patients at higher risk.

## Limitations

We investigated a relatively small cohort of low-risk patients with overall good outcomes. The results may be different in high-risk patients. Our study was not designed to reveal the mechanism behind the correlation between the reduced elevation of cfDNA and worst outcomes. Therefore, the explanations for this phenomenon are, in part, speculative and based on other studies. Euroscore and STS scores were designed to predict mortality. Since we had no mortalities, we compared the performance of these scores to predict different outcomes.

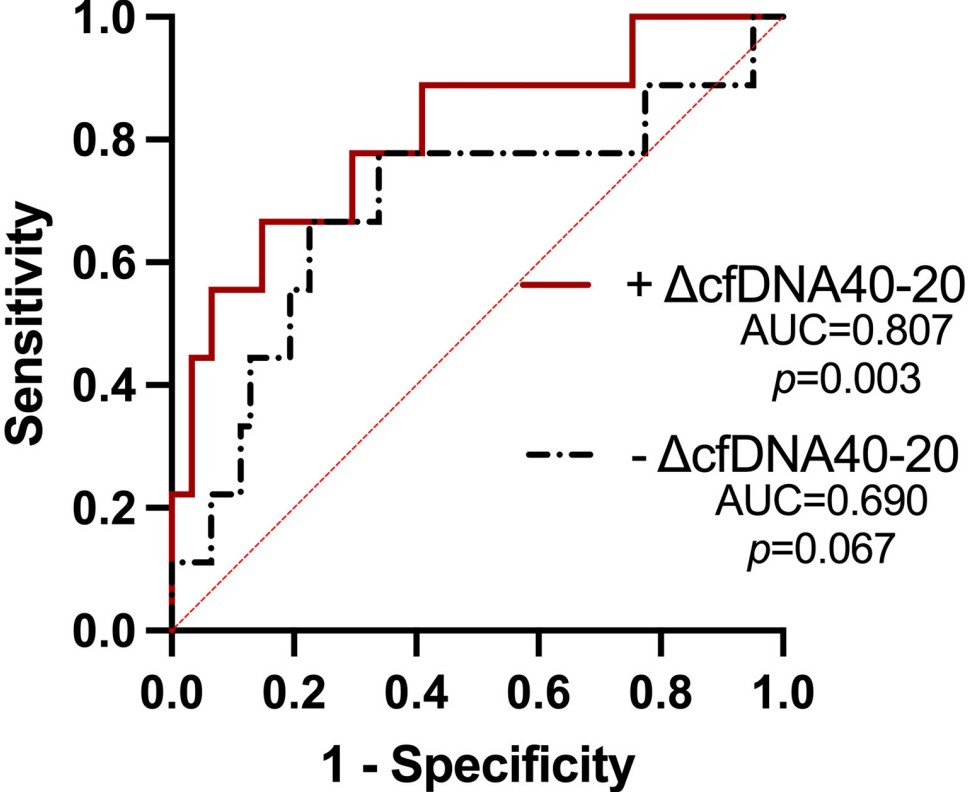

**Fig 4. Logistic regression models for prolonged hospital LOS for ΔcfDNA 40–20 adjusted to time on CPB.**
Multivariate ROC curve of long hospital LOS (>15 days) according to logistic regression models that include ΔcfDNA 40–20 and time on CPB.

## Conclusion

This is the first study to characterize the dynamic changes in cfDNA on-pump cardiac surgery. cfDNA levels increase with a peak at cross-clamp removal. Reduced cfDNA elevation between 20 to 40 minutes on CPB (ΔcfDNA 40–20) was markedly associated with PVD and worse outcomes. Possibly, this negative correlation is related to neutrophils dysfunction or is the reflection of microvasculature occlusion that prevents the degradation of NETs to circulating cfDNA. The simple fluorescent assay we used to measure cfDNA may enable real-time monitoring of ΔcfDNA 40–20 to detect patients at higher risk. Further studies on the mechanism behind the negative association of ΔcfDNA 40–20 with PVD and outcomes are warranted.

## Acknowledgments

The authors thank Valeria Frishman for her excellent technical assistance.

## Author Contributions

**Conceptualization:** Shlomo Yaron Ishay, Sahar Gideon, Amos Douvdevani.

**Data curation:** Shlomo Yaron Ishay, Muhammad Abu-Tailakh, Lior Raichel, Tal F. Hershenhoren, Menahem Matsa, Oren Lev-Ran, Sahar Gideon.

**Formal analysis:** Shlomo Yaron Ishay, Muhammad Abu-Tailakh, Amos Douvdevani.

**Funding acquisition:** Amos Douvdevani.

**Investigation:** Shlomo Yaron Ishay, Sahar Gideon.

**Methodology:** Sahar Gideon, Amos Douvdevani.

**Project administration:** Shlomo Yaron Ishay, Lior Raichel, Tal F. Hershenhoren, Amos Douvdevani.

**Resources:** Sahar Gideon, Amos Douvdevani.

**Supervision:** Amos Douvdevani.

**Writing – original draft:** Muhammad Abu-Tailakh, Lior Raichel, Menahem Matsa, Oren Lev-Ran, Sahar Gideon, Amos Douvdevani.

**Writing – review & editing:** Shlomo Yaron Ishay, Amos Douvdevani.

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
