## [Decision Letter · Decision Letter 0]

18 May 2022

PONE-D-22-11026A prospective cohort study of dynamic cell-free DNA elevation during cardiac surgery with cardiopulmonary bypassPLOS ONE

Dear Dr. Douvdevani,

Thank you for submitting your manuscript to PLOS ONE. After careful consideration, we feel that it has merit but does not fully meet PLOS ONE’s publication criteria as it currently stands. Therefore, we invite you to submit a revised version of the manuscript that addresses the points raised during the review process.

The reviewers noted a number of areas that need to be addressed before I believe this paper can be published including some more information regarding patient selection, consideration of factors other than NETosis driving the changes seen as well as rephasing of certain sections to ensure the findings of the manuscript are not overstated.

We look forward to receiving your revised manuscript.

Kind regards,

Daniel M. Johnson, PhD

Academic Editor

PLOS ONE

Journal Requirements:

"This work was supported by the Dr. Montague Robin Fleisher Kidney Transplant Unit Fund."

"No"

"NO authors have competing interests"

6. We note that you have indicated that data from this study are available upon request. PLOS only allows data to be available upon request if there are legal or ethical restrictions on sharing data publicly. For more information on unacceptable data access restrictions, please see http://journals.plos.org/plosone/s/data-availability#loc-unacceptable-data-access-restrictions. 

Reviewers' comments:

Reviewer's Responses to Questions

**Comments to the Author**

1. Is the manuscript technically sound, and do the data support the conclusions?

Reviewer #1: Partly

Reviewer #2: Partly

2. Has the statistical analysis been performed appropriately and rigorously? 

Reviewer #1: N/A

Reviewer #2: Yes

3. Have the authors made all data underlying the findings in their manuscript fully available?

Reviewer #1: No

Reviewer #2: Yes

4. Is the manuscript presented in an intelligible fashion and written in standard English?

Reviewer #1: Yes

Reviewer #2: Yes

5. Review Comments to the Author

Reviewer #1: The present manuscript describes the predictive value of cfDNA increase for the outcome of patients undergoing cardiac surgery with CPB. The authors found that high differences in cfDNA values 20 min and 40 min after CPB initiation may be prognostic for better recovery, reduced LOS at ICU and in hospital. This study is important and may be considered for publication, after the authors have addressed the following remarks and questions:

- It is noteworthy that plasma cfDNA levels do not solely reflect NETosis but may rather arise from the damaged tissue (e.g. endothelial injury, cardiac injury). Thus, higher neutrophil numbers might be due to an increased inflammatory response. This aspect should be considered by the authors. Hence, the conclusion in the abstract section (lines 38-39) is not supported by the data in the manuscript, as the authors did not distinguish between cfDNA and NETs.

- Line 55: NETs generally induce thrombosis and endothelial injury, independent of degradation

- Detailed description of the method used to quantifiy cfDNA should be included in the Methods section on page 5, including the type of anticoagulant used, centrifugation steps, Sybr gold concentration etc. Please also include manufacturers’ name.

- Which DNA standard was used in this study? Range of DNA standard?

- Include the type of statistical test in all figure captions

- Lines 175-176: did patients with PVD have lower baseline cfDNA levels? If so, the authors should include this information in the manuscript

- Did patients with high delta cfDNA40-20 levels (Fig1B) have higher cfDNA levels at baseline? Please include patients’ cfDNA baseline levels in the individual groups (Fig 1B). Are there any statistical differences at the time point 20 min CPB between the different groups? It looks like the cfDNA levels in the first group are markedly elevated when compared to the groups of patients with longer ICU stays (Fig 1B). Same for the time point 40 min.

- In fact, high delta cfDNA40-20 negatively correlates with ICU / hospital LOS. How do the authors explain that? The assumption (lines 254-255) that inverted correlation of delta cfDNA40-20 and neutrophils with LOS may be explained by the elimination of NETing neutrophils from the circulation due to microvascular obstruction is very speculative and not satisfactory. Does rather a strong initial inflammatory response (reflected by high neutrophil counts, high cfDNA levels) support patients’ recovery? Did the authors measure additional inflammatory markers like IL-6? As already mentioned, it is mandatory to prove if 1. cfDNA levels differ at baseline between groups and 2. cfDNA levels at 20 min and also at 40 min are different between groups (Fig 1 B)

- Although this study did not aim to explain the mechanism behind these findings, I strongly recommend to include additional data showing dynamics of at least one inflammatory marker (e.g. CRP, IL-6) that may help to interpret these unexpected results.

Reviewer #2: “A prospective cohort study of dynamic cell-free DNA elevation during cardiac surgery with cardiopulmonary bypass” is a prospective cohort study by Shlomo Y Ishay and colleagues who ascertained the trajectory of cell-free DNA (cfDNA) in patients undergoing cardiopulmonary bypass (CPB) for elective coronary artery bypass grafting (CABG), and its association with outcomes such as prolonged in-hospital or intensive care unit (ICU) length of stay (LOS).

The dynamic assessment of cfDNA throughout CPB is interesting and may provide insights on the early acute-phase reaction that leads to variable degrees of postoperative systemic inflammatory response in cardiac surgery patients.

The acute reaction to cardiac surgery and CPB is indeed a very interesting topic that deserves further investigations, so the authors need to be commended on this effort. However, the immune-inflammatory response is multifaceted and intricate to the point that drawing conclusions is just a daunting task and many pieces of research result controversial. Moreover, the narrative of the inflammatory reaction to CPB surgery (i.e., abstract and introduction section) sounds somewhat old-fashioned and part of the referenced bibliography should be replaced with more recent papers.

Overall, unfortunately this paper misses the criteria for publication. My points are as follows.

METHODS

The methods section is confusing, and some fundamental clarifications are needed.

The authors present data from a cohort of 71 consecutive elective and low-risk patients who underwent CPB-assisted CABG more than a decade ago. Can the author clarify the choice of this very specific cohort numerosity and why they are submitting the manuscript more than 10 years after the study execution?

Primary endpoints were:

- Assess the kinetic of cfDNA during CPB;

- Ascertain its association with prolonged ICU and in-hospital stay;

- Evaluate its impact on surgery-related death.

Secondary endpoints were:

- Investigate the association of the cfDNA trajectories with routine lab data;

- Compare the prognostic value of the cfDNA trajectories with standard risk score (STS and EuroSCORE II).

The definition of some outcomes is unclear. Indeed, a prolonged ICU-LOS is defined as >48h (line 84, table 1), but also as >18h (lines 108-109, clinicaltrials.gov) whereas a prolonged in-hospital stay is defined as >15 days (line 85, table 1), but also as >7 days (lines 108-109, clinicaltrials.gov). Moreover, the mechanical ventilation time is mentioned neither as primary nor as secondary outcome but the extubation protocol was thoroughly described and further analysis was provided. Should we consider mechanical ventilation time as one of the outcomes of interest?

Furthermore, the section entitled “Criteria for ICU and hospital discharge” did not provide any information about the actual criteria for discharge neither from ICU nor from the surgical ward instead it merely defined what was considered to be normal and prolonged ICU- and hospital-stay. Disappointingly, these definitions were incoherent with the rest of the manuscript.

Another major issue is represented by the long-term follow-up (i.e., 54 months) that, in light of the analysis and results provided, sounds very controversial to me, and that basically dampened the modernity of the early-outcome data without providing any relevant information.

Minor issues:

- Lines 75-76: “SUMC is the sole tertiary medical center in the Negev region” is an irrelevant statement and should be deleted.

- Line 91: “Ascending aorta-to-right atrial appendage” should be rephrased as “right atrial appendage-to-ascending aorta” to better describe the direction of the extracorporeal circulation.

RESULTS

There are also some major areas of concern with this section.

The first concern is related to the choice of the delta-cfDNA 40-20 min. The concentration of cfDNA increased early at CPB-onset, peaked at cross clamp removal in a time-dependent manner, and returned to baseline 30 minutes after chest closure. The authors need to clarify why the delta-cfDNA baseline-to-peak did not undergo analysis whereas the delta-cfDNA 40-20 min was rather chosen.

Furthermore, the results section starts with the description of the cohort characteristics and of the early outcomes. This is absolutely correct. However, after this preface, the results of the primary outcomes analysis should have been reported first, followed by those of the secondary outcomes. In this sense, the results section appears chaotic, and rephrasing the whole section is mandatory.

Primary endpoints:

- The kinetic of cfDNA is well described;

- Its association with the ICU and in-hospital stay is also well described;

- Instead, the analysis of its impact of 30-days surgery-related death in such a small group of very low-risk elective patients is somehow nonsense.

Secondary endpoints:

- Early postoperative lab data showed the reduction of parameters like hemoglobin, albumin, and calcium (lines 133-134): this phenomenon is trivial, expected, and it is due to several factors such as hemodilution, blood loss, adhesion to the CPB materials (i.e., albumin), and hemostasis, all definitely not being related to cfDNA. This specific analysis is speculative and perhaps the evaluation of the potential associations of cfDNA levels with those of other common inflammatory markers (e.g., C-reactive protein, lactate dehydrogenase, etc.) would have represented a more consistent point. Indeed, all this was not mentioned in the discussion section. Moreover, the postoperative finding of neutrophilia and lymphopenia is a common occurrence in cardiac surgery patients, and it is considered itself as a marker of inflammatory activation. Its correlation with the cfDNA trajectory does not implicate a causation (notwithstanding, I understand rational behind the possible explanation that was provided in the discussion section).

- The prognostic value of the cfDNA kinetic for the indexed outcomes appears satisfactory as compared to those of the conventional scores. However, it must be noted that these scores were designed and validated to predict early mortality (0% at 30 days), not to predict ICU or in-hospital stay. Hence, this finding can be considered somehow illegitimate.

- The analysis of the predictive performance of the delta-cfDNA 20-40 min for mortality at follow-up was not provided. The low incidence of late mortality is an explanation. However, it is difficult to me to understand the rational behind this investigation given the low-risk cohort and the fact that the hypothetical comparison with prognostic tools calibrated for early mortality would have add almost no information to the manuscript.

DISCUSSION

As I previously stated, the postoperative inflammatory response is complex to the point that drawing conclusions is a daunting task. Indeed, most of the discussion is based on hypothesis and possible explanations. The only point is that a worse outcome would be expected with a sharp rise in the levels of a DAMP like cfDNA, and this was not the finding of this research. NET-osis and microcirculatory angiopathy are indeed a possible explanation. The correlation with peripheral vascular disease is interesting, but a clear definition of PVD is needed.

However, I think that the finding of this work just shows how the intraoperative activation of immune-inflammatory pathways represents a protective mechanism, that can be maladaptive if dysregulated.

The analysis of this study in the context of the existent evidence is well performed and I agree that a true comparison with the result of other works is unreliable because of the different populations and timeframes. Finally, the limitations section appears adequate.

I offer this criticism with respect and appreciate the authors' time very much.

6. PLOS authors have the option to publish the peer review history of their article (what does this mean?). If published, this will include your full peer review and any attached files.

Reviewer #1: No

Reviewer #2: **Yes: **Domenico Paparella

---

## [Author Response · Author response to Decision Letter 0]

28 Jul 2022

Our detailed response to reviewers' and Editor's comments is attached in the "response to reviewers" File.

---

## [Decision Letter · Decision Letter 1]

7 Oct 2022

A prospective cohort study of dynamic cell-free DNA elevation during cardiac surgery with cardiopulmonary bypass

PONE-D-22-11026R1

Dear Dr. Douvdevani,

We’re pleased to inform you that your manuscript has been judged scientifically suitable for publication and will be formally accepted for publication once it meets all outstanding technical requirements. Apologies, once again, for the delay in getting this decision to you

Kind regards,

Daniel M. Johnson, PhD

Academic Editor

PLOS ONE

Additional Editor Comments (optional):

Reviewers' comments:

Reviewer's Responses to Questions

**Comments to the Author**

1. If the authors have adequately addressed your comments raised in a previous round of review and you feel that this manuscript is now acceptable for publication, you may indicate that here to bypass the “Comments to the Author” section, enter your conflict of interest statement in the “Confidential to Editor” section, and submit your "Accept" recommendation.

Reviewer #1: All comments have been addressed

2. Is the manuscript technically sound, and do the data support the conclusions?

Reviewer #1: Partly

3. Has the statistical analysis been performed appropriately and rigorously? 

Reviewer #1: Yes

4. Have the authors made all data underlying the findings in their manuscript fully available?

Reviewer #1: Yes

5. Is the manuscript presented in an intelligible fashion and written in standard English?

Reviewer #1: (No Response)

6. Review Comments to the Author

Reviewer #1: The authors have addressed almost all the points I raised and made necessary changes to the manuscript. Unfortunately, interpretation of results is still speculative, because the authors were not able to provide additional data, which might help to explain the relationship between cfDNA, inflammation and patients’ outcome.

7. PLOS authors have the option to publish the peer review history of their article (what does this mean?). If published, this will include your full peer review and any attached files.

Reviewer #1: No

---

## [Editor Report · Acceptance letter]

20 Oct 2022

PONE-D-22-11026R1 

A prospective cohort study of dynamic cell-free DNA elevation during cardiac surgery with cardiopulmonary bypass 

Dear Dr. Douvdevani:

I'm pleased to inform you that your manuscript has been deemed suitable for publication in PLOS ONE. Congratulations! Your manuscript is now with our production department. 

Kind regards, 

on behalf of

Dr. Daniel M. Johnson 

Academic Editor

PLOS ONE